# Comparative Analysis of the Relative Fragmentation Stabilities of Polymorphic Alpha-Synuclein Amyloid Fibrils

**DOI:** 10.3390/biom12050630

**Published:** 2022-04-25

**Authors:** Sarina Sanami, Tracey J. Purton, David P. Smith, Mick F. Tuite, Wei-Feng Xue

**Affiliations:** 1Kent Fungal Group, School of Biosciences, Division of Natural Sciences, University of Kent, Canterbury CT2 7NJ, UK; ss2409@kent.ac.uk (S.S.); t.j.purton@kent.ac.uk (T.J.P.); m.f.tuite@kent.ac.uk (M.F.T.); 2Biomolecular Research Centre, Sheffield Hallam University, Sheffield S1 1WB, UK; d.p.smith@shu.ac.uk

**Keywords:** atomic force microscopy, amyloid, fibril fragmentation, fibril division, stability, image analysis, sonication

## Abstract

The division of amyloid fibril particles through fragmentation is implicated in the progression of human neurodegenerative disorders such as Parkinson’s disease. Fragmentation of amyloid fibrils plays a crucial role in the propagation of the amyloid state encoded in their three-dimensional structures and may have an important role in the spreading of potentially pathological properties and phenotypes in amyloid-associated diseases. However, despite the mechanistic importance of fibril fragmentation, the relative stabilities of different types or different polymorphs of amyloid fibrils toward fragmentation remain to be quantified. We have previously developed an approach to compare the relative stabilities of different types of amyloid fibrils toward fragmentation. In this study, we show that controlled sonication, a widely used method of mechanical perturbation for amyloid seed generation, can be used as a form of mechanical perturbation for rapid comparative assessment of the relative fragmentation stabilities of different amyloid fibril structures. This approach is applied to assess the relative fragmentation stabilities of amyloid formed in vitro from wild type (WT) α-synuclein and two familial mutant variants of α-synuclein (A30P and A53T) that generate morphologically different fibril structures. Our results demonstrate that the fibril fragmentation stabilities of these different α-synuclein fibril polymorphs are all highly length dependent but distinct, with both A30P and A53T α-synuclein fibrils displaying increased resistance towards sonication-induced fibril fragmentation compared with WT α-synuclein fibrils. These conclusions show that fragmentation stabilities of different amyloid fibril polymorph structures can be diverse and suggest that the approach we report here will be useful in comparing the relative stabilities of amyloid fibril types or fibril polymorphs toward fragmentation under different biological conditions.

## 1. Introduction

Amyloid fibrils are proteinaceous polymers characterized by their cross-β core molecular structure. The non-covalent intermolecular forces within the cross-β amyloid cores running parallel to the fibril axis, from the β-strands that are oriented perpendicularly to the fibril axis, stabilises the amyloid structure, making them strong and biochemically resilient assemblies [1,2]. The formation of amyloid structures is observed in association with many human neurodegenerative diseases. For example, assembly of the α-synuclein protein into amyloid aggregates is associated with Parkinson’s disease (PD), Lewy body dementia (LBD), multiple system atrophy (MSA), and other synucleinopathies. Recent advances in structural studies of amyloid fibrils using solid-state nuclear magnetic resonance, electron diffraction, X-ray diffraction and cryo-electron microscopy (cryo-EM) have extended our understanding of the complexities of amyloid fibril structures in relation to pathology. It is now apparent that, like other amyloid-forming sequences, α-synuclein amyloid fibrils can adopt different and distinct characteristic structures, in a phenomenon called structural polymorphism (e.g., [3,4]). How these structural polymorphisms link to distinct histopathological phenotypes remains to be established [4,5,6]. 

Several point-mutations in the α-synuclein gene (*SNCA*) have been linked to familial PD [7,8] with some of these (e.g., A30P [9] and A53T [10]) being associated with early-onset PD. However, no precise molecular and mechanistic explanation exists on how these familiar point mutations and the effects of these mutations on the α-synuclein aggregation process and their fibril structures [11,12] subsequently affect the aetiology of associated disease conditions [6]. Studies to date suggest that α-synuclein amyloid fibrils formed from these familiar mutation variants are morphologically different to those formed from WT α-synuclein and the distinct polymorphs they form may comprise structural differences that can affect seeding, aggregation, and interactions with lipid membranes [11,12]. 

Amyloid formation follows a nucleation-dependent polymerisation mechanism [13,14], where soluble protein precursors adopt the amyloid state through a slow and rate-limiting primary nucleation process. The aggregates subsequently grow through fibril elongation, for example, by the addition of monomers at fibril ends [15], and they propagate exponentially through secondary nucleation [16] and fibril fragmentation [17,18]. Thus, fibril fragmentation is a key process in which fibrils divide, leading to the production of small fibril seeds and the acceleration of amyloid growth and propagation. In vivo, fibril fragmentation can occur through catalysis by molecular chaperones such as Hsp104, Hsp70, and Hsp40 [19]. The relative fragmentation stability differences of distinct α-synuclein fibril polymorphs revealed by sonication have previously been documented [20] but were not quantified. These differences could contribute to differences in disease aetiology due to their relation to the aggregation processes and their sensitivity to solution conditions, particularly for polymorphic amyloid-forming proteins such as α-synuclein [20,21,22].

Here, we report a comparative analysis of the dynamic fragmentation stability of morphologically distinct fibrils formed in vitro from WT α-synuclein and two of its familiar single-point mutation variants (A30P, and A53T). The relative resistance of the three types of α-synuclein amyloid fibrils toward fragmentation promoted by controlled sonication was determined. We find that the sonication-induced fragmentation of all three α-synuclein amyloid fibril types is length dependent, but their overall stability towards sonication-induced fragmentation is markedly different, confirming the possibility of differential fragmentation stabilities for different structural polymorphs of amyloid fibrils. Our results also show that A30P and A53T α-synuclein fibrils display more resilience towards fragmentation than their WT counterparts. These differences in fragmentation stability could be linked to the propensity of A30P and A53T to form fibril clusters, highlighting the surfaces properties of different fibril polymorphs as a key physicochemical property that may provide a link between the polymorphic structures of amyloid to their biological and pathological roles.

## 2. Materials and Methods

### 2.1. Recombinant Expression and Purification of Monomeric α-Synuclein

Recombinant human WT, A30P, and A53T α-synuclein proteins were produced and purified according to the method used by Cappai et al. [23], with the addition of a stepped ammonium sulphate precipitation (30% to 50%) step prior to anion exchange chromatography. Briefly, for each α-synuclein sample, BL21 DE3 *E. coli* cells were transformed with the pET23a plasmid containing the WT human α-synuclein sequence or its point-mutation variants. The cells were cultured in LB, supplemented with ampicillin sodium salt (100 μg/mL). After overnight incubation at 37 °C, the pellets were resuspended in LB with the same ampicillin concentration. IPTG was added to the culture when its OD600 reached about 0.6 to induce recombinant α-synuclein expression. After 4 h of incubation, cells were harvested by centrifugation and resuspended in 20 mL of lysis buffer (20 mM Tris, 100 mM NaCl, pH 7.5) and complete protease inhibitor tablets were added to the cells in the spun down pellet. The cells were sonicated on ice and cell debris was then removed by centrifugation at 13,000 rpm for 30 min. The resulting supernatant was acidified to pH 3.5 using 1 M HCl and incubated on a tube roller at 4 °C for 30 min. The pH was then neutralised to 7.5 using NaOH on ice. The solution was subsequently treated with 30% (NH_4_)_2_SO_4_ and incubated at 4 °C for 20 min on a tube roller. The solution was then centrifuged at 13,000 rpm and the supernatant was treated with 50% (NH_4_)_2_SO_4_ and incubated at 4 °C on a tube roller for another 20 min. Further centrifugation at 13,000 rpm for 30 min was performed and the resulting pellet was dissolved in 20 mM Tris, pH 7.5, to a final volume of 2.5 mL for anion exchange chromatography with a HiTrapTM Q FF column as otherwise previously described [23].

### 2.2. In Vitro Assembly of α-Synuclein Amyloid Fibrils

WT, A30P and A53T α-synuclein amyloid fibril samples were formed by buffer exchanging purified protein monomers into fibril-forming buffer (FFB, 10 mM sodium phosphate buffer, pH 7.4) using a PD-10 column with Sephadex G-25 medium (GE Healthcare). Protein concentration was determined by absorbance at 280 nm, and the sample solutions were diluted to 50 μM monomer equivalent concentration with FFB. The samples were subsequently incubated at 37 °C in a shaking incubator with agitation set at 180 rpm for at least 14 days.

### 2.3. Fibril Fragmentation by Controlled Sonication

Fragmentation of WT, A30P and A53T α-synuclein amyloid fibrils was carried out by sonication using a Qsonica Q125 20 kHz probe sonicator with a 2 mm probe and set at 20% amplitude (lowest possible amplitude setting). For each α-synuclein fibril sample, 400 μL of an initial sample (50 uM monomer equivalent concentration for all three fibril samples) in a LoBind Eppendorf tube was placed on ice and pulse sonicated in 5 s intervals, with 5 s of rest between the pulses to counteract the heating of the solution. This was performed in multiple cycles to attain the required time duration in the sonication time-point sequence, yielding total durations of 5, 10, 20, 40, 80, 320 and 640 s of mechanical perturbation by sonication.

### 2.4. AFM Imaging and Image Analysis

For samples sonicated 10 s or longer, used for quantitative analysis of the fragmentation reactions, after each sonication duration time point, the sonicated samples were immediately diluted into the FFB to a final concentration of 1 μM (WT), 5 μM (A30P) or 25 μM (A53T) to be immediately used for AFM imaging so that sufficient coverage of evenly dispersed and well-separated fibril particles could be obtained on images. For non-sonicated samples or samples sonicated for 5 s, only the A30P fibril sample was diluted to a final concentration of 25 μM, while WT and A53T α-synuclein fibril samples were used without further dilution for AFM imaging. Samples were prepared for AFM imaging by incubating a 20 μL fibril sample solution on a freshly cleaved mica surface for 5 min at room temperature. The mica surface was subsequently washed with 1 mL of stereo-filtered Milli-Q water (prepared using a Minisart Syringe Filter with 0.2 μm pores) to remove non-deposited fibrils. The sample on mica support was then gently dried with nitrogen flow. AFM imaging was carried out on a Bruker Multimode 8 scanning probe microscope with a Nanoscope V controller, using the ScanAsyst peak-force tapping imaging mode. Bruker ScanAsyst-Air probes (silicon nitride tip with a nominal tip radius of 2 nm, nominal spring constant 0.4 N/m, and nominal resonant frequency 70 kHz) were used throughout. Images were captured at 10 × 10 μm or 4 × 4 μm scan sizes with a fixed pixel resolution of 2048 × 2048 pixels. The images were processed using the supplied software (NanoScope Analysis 1.5, Bruker, Karlsruhe, Germany) to remove scanner tilt and bow. The images were then imported into MATLAB, where the length and height of individual fibril particles were measured. The fibril length and height distributions were obtained as previously described [24,25]. For the fibril length distributions, length-dependent bias in the deposition step for imaging or during the fibril tracing step of image analysis was taken into account as previously described [24]. A model of pure fragmentation was finally fit to the time-dependent length decay data as previously described (see [26,27] and the tutorial_DETX MATLAB code examples deposited on Github: https://github.com/mtournus/Fragmentation accessed on 23 August 2021). Briefly, assuming nucleation, elongation, and depolymerisation processes are negligible in their rates compared to fibril fragmentation during the sonication procedure, the *γ* parameter describing how the fragmentation rate constants depend on fibril length can be extracted by fitting the following equation to the time (*t*) dependence of average fibril lengths *µ*.
(1)$$\{\begin{array}{c} \mu \left(t\right)=C\cdot t^{-1/\gamma };\;\;\;t>t_{s}\;\\ \;\mu \left(t\right)=C\cdot t_{s}{}^{-1/\gamma };\;\;\;t\leq t_{s}\; \end{array}$$

In Equation (1), *t_s_* is the time point from which the best-fit pure fragmentation model is likely reliable in describing the experimentally observed length decay and *C* is a constant. The *α* parameter describing the typical magnitude of the division rate constants was then calculated with *γ* obtained from Equation (1) using the following equation:(2)$$\alpha ={\left(\frac{1}{\gamma }\frac{t^{-1}}{\int_{0}^{\infty }x^{\gamma }f\left(t,\;x\right)dx}\right)}^{\frac{1}{\gamma }}$$

In Equation (2), *x* is fibril length and *f*(*t,x*) is the normalised length distribution at time *t*. The *α* parameter was obtained by averaging the *α* values obtained for each *t > t_s_* time-points. The fragmentation rate constants for any fibril length are then *B*(*x*) *=* (*α·x*)*^γ^ s*^−1^. Parameter standard errors were estimated by the Jackknife resampling method.

## 3. Results

### 3.1. Amyloid Fibrils Formed from A30P and A53T α-Synuclein Display Distinct Polymorphic Structures Compared to WT α-Synuclein Fibrils

To investigate the impact of the disease-associated familiar point mutations A30P and A53T on the stability of α-synuclein amyloid towards fragmentation, WT, A30P, and A53T α-synuclein monomers were first assembled in vitro into amyloid fibrils under identical solution conditions at neutral pH (10 mM sodium phosphate buffer at pH 7.4). Under these solution conditions, twisted filaments morphologically typical for amyloid fibrils (Figure 1) were formed after two weeks’ incubation under gentle agitation in a shaker incubator at 180 rpm and 37 °C. Nano-scale imaging by atomic force microscopy (AFM) confirmed that all three α-synuclein sequences formed long, straight, and twisted fibrils that are similar to α-synuclein amyloid fibrils previously imaged by AFM [12,21,26,28]. Interestingly, all three amyloid fibril samples displayed considerable clustering of fibrils, with the A53T sample resulting in the most visibly apparent tangling and bundling of fibrils.

Amyloid fibrils formed from α-synuclein sequences in vivo and in vitro can adopt a wide variety of different structures due to the phenomenon known as structural polymorphism [29]. Structural analysis of individual fibrils [30,31] observed in the topographical AFM image data (Figure 1c,f,i) revealed that the three α-synuclein sequences each resulted in amyloid fibrils of distinctively different dominant morphologies. Digitally straightened images and reconstructed three-dimensional (3D) surface envelope models of individual fibrils formed from each of the α-synuclein variants (Figure 1c,f,i, respectively) revealed left-hand twisted fibrils in all three cases. However, the fibril width and twist patterns, as seen in the 3D surface envelopes and the height profiles, were notably different when comparing the α-synuclein WT, A30P and A53T amyloid fibrils. A30P fibrils appeared generally thinner compared to WT α-synuclein fibrils, with the AFM height profile of A30P fibrils showing an around 1 nm lower height compared to WT (Figure 1f compared to c). On the other hand, A53T fibrils had a more elongated cross-section compared to WT, showing a larger difference between maxima and minima in the height profile compared to the WT (Figure 1i compared to c). Importantly, these results demonstrated that the three α-synuclein sequences each form different dominant fibril polymorphs, which can be readily distinguished structurally by AFM image analysis of their height distributions.

### 3.2. Controlled Sonication Promotes Rapid Fragmentation of α-Synuclein Amyloid Fibrils

The dynamic stability of WT, A30P and A53T α-synuclein fibrils towards fragmentation was next probed by controlled sonication. We have previously employed stirring as the mechanical perturbation used to probe the relative fragmentation stability of different types of amyloid fibrils [26]. Controlled sonication is a method commonly used to rapidly fragment amyloid fibrils and for reproducibly generating amyloid seeds [32,33,34]. Consequently, we employed controlled sonication as a rapid mechanical perturbation method to analyse the relative fragmentation stability of the three α-synuclein amyloid types. The three α-synuclein amyloid samples were treated under identical sonication regimes (see Methods section) for a period of time up to 640 s. Typical 2048 × 2048 pixel AFM height images covering 10 × 10 µm surface areas with fibril particles produced by the controlled sonication treatment of the samples for different lengths of time are displayed in Figure 2.

As seen in Figure 2, after 5 s of controlled sonication, all three types of α-synuclein amyloid fibrils (WT, A30P and A53T) were effectively dispersed and populations of well-separated shorter fibril particles started to appear, as expected. Further sonication, up to 640 s, resulted in rapid and effective fragmentation of all three fibril types, which increased the number of small, short fibril particles. As sonication progressed for each of the three α-synuclein fibril types, short fibril particles with apparently similar morphological appearances and fibril widths were generated compared with non-fragmented fibrils of the same type, suggesting that the sonication process did not alter the morphological structural characteristics of the fibrils as they divide. Interestingly, the WT α-synuclein amyloid particles at the later time points remained dispersed compared to 5–10 s sonication time points, while A30P and A53T fibrils particles showed a tendency to remain clustered together at the later time point compared to the 5–10 s sonication time points. This behaviour suggested differences in the surface properties of the three α-synuclein fibril types, and that the surface properties of each fibril type persisted through the sonication process.

### 3.3. Quantitative Image Analysis of α-Synuclein Amyloid Fibrils Undergoing Fragmentation by Controlled Sonication Show Polymorph-Dependent Length Distribution Decays

Quantitative measurements of the length distributions and the height distributions of WT, A30P, and A53T α-synuclein fibril particles undergoing division by fragmentation promoted by controlled sonication and monitored by AFM (Figure 2) were next carried out (Figure 3). As expected [26,35], the average length of the fibril particles was reduced and an increasing number of smaller particles were produced as the length of time the samples were sonicated was increased. However, as seen in Figure 3, the average length of the WT α-synuclein amyloid particles reduced to 56.9 nm, while for A30P α-synuclein fibrils, the average length reduced to 71.0 nm, and for A53T α-synuclein fibrils, the average length reduced to 117.8 nm after the same 640 s of controlled sonication. These findings suggested that both A30P and A53T α-synuclein amyloid fibrils are more resistant to fragmentation by the mechanical perturbation provided by the sonication treatment, since the length distributions at long perturbation time points are indicative of the dynamic stability of fibrils towards fragmentation under the conditions applied [26,27]. In terms of the height distributions indicative of the distributions of fibril polymorphs and widths, the controlled sonication did not change the overall shape and location of the height distributions throughout the duration of the sonication treatment for all three samples (Figure 3c,d, Table 1). The average height for the WT α-synuclein fibrils remained around 7.8 nm throughout, and for the A30P sample, around 6.4 nm. For the A53T sample, the average height was around 8.2 nm, although the A53T α-synuclein fibrils exhibited a wider range of heights compared with the other two samples. Interestingly, the height distributions of A30P and A53T α-synuclein fibril samples showed pronounced tails at heights taller than 8 nm. The tails of the height distributions were most pronounced for the A53T followed by A30P α-synuclein fibril samples and remained consistent throughout the controlled sonication treatment. This observation is consistent with the tendency for A53T, followed by A30P α-synuclein fibrils, to form clusters, which may be indicative of their ‘sticky’ surface properties. Thus, the quantitation of the AFM image data showed that controlled sonication promoted rapid decays of fibril lengths but did not notably affect the fibril width distributions or the distributions of fibril polymorphs for each of the three α-synuclein variant fibril samples. The results also showed that the sonication treatment did not cause the same rates of length distribution decay for the three fibril samples, suggesting detectable differences in how the different α-synuclein fibril polymorphs resisted the mechanical perturbation provided by the sonication treatment applied.

### 3.4. The Rates of Fibril Fragmentation Promoted by Controlled Sonication Are Distinct for the Different Fibril Polymorphs Formed from WT, A30P and A53T α-Synuclein Variants

To confirm the apparent differences in the resistance towards sonication-induced fibril fragmentation for the WT, A30P and A53T α-synuclein amyloid fibril samples, we next analysed the division rates of the sonication-induced fibril fragmentation processes. By applying the model of pure fragmentation we previously developed [26,27], the fibril length (*x*)-dependent division rate constants, *B*(*x*), as well as the parameters *γ* describing how the rate constants depend on fibril length, and *α* describing the typical magnitude of the division rate constants, were directly extracted from the data. We first determined the *γ* value associated with the fragmentation of each of the α-synuclein fibril types (Table 1) by fitting a power law to the average lengths as a function of sonication duration [26,27] for each of the three α-synuclein datasets. This is visualised as straight lines in the log–log plot of the average lengths versus sonication duration (Figure 4a). As seen in Table 1 and Figure 4a, both A30P and A53T α-synuclein fibrils displayed higher *γ* values compared with WT. This indicates that the fibril fragmentation rates for A30P and A53T α-synuclein fibrils are more strongly length dependent compared to WT. As consequence, the decay of A30P and A53T α-synuclein fibril lengths progressed slower compared to WT α-synuclein fibrils under the sonication conditions applied. 

Finally, we determined the *α* values and compared the fragmentation rate constants *B*(*x*) *=* (*α·x*)*^γ^* s^−1^ [26,27] for the three α-synuclein fibrils types (Table 1, Figure 4b). As seen in Table 1 and Figure 4b, the fragmentation rate constants *B*(*x*) for A30P and A53T α-synuclein fibrils are at least one order of magnitude smaller compared to WT fibrils for fibrils 100 nm in length (*x*) or shorter. For fibrils 20 nm in length or shorter, the differences in *B*(*x*) increased to more than two orders of magnitude. Thus, for fibrils between ~10 to ~1000 nm in length that were observed in the AFM experiments, A53T α-synuclein fibrils are most resistant to fragmentation followed by A30P and WT α-synuclein fibrils, which are least resistant to fragmentation (i.e., in Figure 4b, the rough rank order from the top, higher rate constants to the bottom lower rate constants is grey, WT, followed by purple, A30P, and pink, A53T, lines) under the solution and sonication conditions applied. The overall picture that emerges from the result is that both A30P and A53T α-synuclein fibrils are more stable towards fibril fragmentation promoted by mechanical perturbation compared to WT α-synuclein fibrils, and this difference in fragmentation stability may arise from their increased surface interactions that promote fibril cluster formation.

## 4. Discussion

Fibril fragmentation is a crucial process in the life cycle of amyloid, as it enables the propagation of the amyloid state by generating small and active fibril seeds, which may be linked with the potential for toxicity [17] and cell-to-cell spreading [32] in the case of disease-associated amyloid. For example, it has been shown that small α-synuclein amyloid fibril fragments 50 nm or less in length show increased seeding and propagation activity compared with their longer counterparts and are the most efficient promoters of the accumulation of phosphorylated α-synuclein in mouse models [22]. Therefore, it is important to assess and compare the relative stabilities of amyloid fibrils toward fragmentation so that the potential of different amyloid types or different polymorphs in generating active amyloid seeds can be understood.

To evaluate the relative fragmentation stability of amyloid fibrils, we have previously developed an experimental approach that involves the fragmentation of pre-formed amyloid samples by mechanical perturbation in the form of stirring using magnetic stirrer bars [26]. This method of mechanical perturbation utilises a rotation action via the magnetic stirrer and is a comparatively gentle method that results in slow rates of fragmentation. Since amyloid fibril fragmentation is a length-dependent process, the small amyloid particles produced as a result of fragmentation are more stable towards further fragmentation compared to their longer parent particles [26,35]. This has the practical experimental ramification that amyloid fragmentation stability experiments have to run over days to weeks, especially since the long-time fragmentation behaviours contain most information on fibril fragmentation rates [26,27]. Sonication has been a commonly used type of mechanical perturbation applied in vitro to promote fibril fragmentation, for example, for reproducible generation of fibril seeds. Sonication is considered to be a vigorous mechanical perturbation in comparison to stirring and promotes rapid fragmentation of amyloid fibrils [33]. Sonication procedures typically involve the use of ultrasonic baths devices [34,36] or probe sonicators [22,32,37]. Here, we have employed controlled sonication using a probe sonicator as the mechanical perturbation in order to probe the relative fragmentation stability of α-synuclein amyloid fibril polymorphs. Sonication provides a stronger, more rapid method of mechanical perturbation than stirring, delivering a discontinuous pulsed mode of action on the fibril samples, and as seen here, resulting in different absolute rate and length dependence of the fragmentation rates when compared with stirring. Nevertheless, the data on the decay of α-synuclein fibril lengths are fully consistent with the predicted behaviours of the pure fragmentation model [26,27]. Relative differences in mechanical fragmentation stability between WT, A30P, and A53T α-synuclein fibrils were also resolved in time-course experiments lasting minutes instead of days. Thus, our results suggest that controlled sonication provides a rapid and useful perturbation method for measuring and comparing the relative fragmentation stabilities of amyloid samples.

To investigate whether different fibril polymorphs could display observable differences in their stabilities towards fibril fragmentation, we compared the relative mechanical fragmentation stability of WT α-synuclein fibrils with fibrils formed from A30P and A53T α-synuclein. Consistent with previous biophysical studies [11,12], these two familiar point-mutation variants of α-synuclein were shown to form amyloid samples dominated by different fibril polymorph structures when compared with WT α-synuclein, i.e., the morphologies of the fibrils formed from WT, A30P and A53T α-synuclein are different from each other (Figure 1). Familiar A30P and A53T α-synuclein point mutations lead to early-onset PD, which could suggest the fibrils formed by these variants have increased susceptibility to fibril fragmentation. Instead, the data here suggest that amyloid formed in vitro from these two α-synuclein variants is more resistant towards fragmentation promoted by the mechanical perturbation employed here. This may be due to increased fibril surface activities of α-synuclein amyloid polymorphs formed from these two variants as A30P and A53T α-synuclein fibril samples seem to exhibit ‘sticky’ surfaces, which increases the likelihood of forming fibril clusters (Figure 2) that could display increased resistance to fragmentation. This explanation is consistent with the observations that A30P and A53T α-synuclein also show increased lipid-induced aggregation and a propensity for surface catalysed secondary nucleation [11]. Thus, A30P and A53T α-synuclein amyloid, once formed in vivo, may be more persistent for cellular disaggregation machineries to clear. Overall, the results demonstrate that the morphological and structural differences between amyloid fibril polymorphs are indeed capable of altering their intrinsic ability to divide by fragmentation, but also suggest that at least some part of the observed differences are meditated by fibril surface interactions. These conclusions support the idea that amyloid surface properties and surface interactions of amyloid structures are key mesoscopic properties to target [38] to better understand the biological impact of amyloid structures and structural polymorphism.

Fibril fragmentation of amyloid fibrils in vivo is dependent on the catalytic activities of molecular chaperones in the cells [19,39]. At the cellular level, aggregate clearance machineries, such as the Hsp40/70/104 chaperone system found in yeast, perform a dual function since their disaggregation activity also mediates the fragmentation of amyloid fibrils and the formation of seeding-competent aggregates [19]. Chaperone-catalysed amyloid fibril fragmentation could increase the overall formation of amyloid aggregates and promote cell-to-cell propagation of the amyloid state, potentially exacerbating disease-related processes [40]. For instance, the role of the chaperone disaggregation machinery in the aggregation and toxicity of α-synuclein aggregates has been studied in vitro [41] and in vivo [42], and points to chaperones as potential amyloid fragmentation mediators. However, under certain conditions, chaperones may also combat aggregation and the toxic effect of α-synuclein amyloid formation [43]. Hence, chaperones may exhibit distinct modes of amyloid disaggregation and fragmentation that are dictated by the morphology and structure of the precise amyloid fibril polymorph the chaperones act upon and the precise environmental conditions during the catalysis of fragmentation reactions. Therefore, the approach we report here could be useful and employed to determine whether individual types of amyloid fibril polymorphs also display distinct propensities for chaperone-catalysed fragmentation in biological milieus.

## Figures and Tables

**Figure 1 biomolecules-12-00630-f001:**
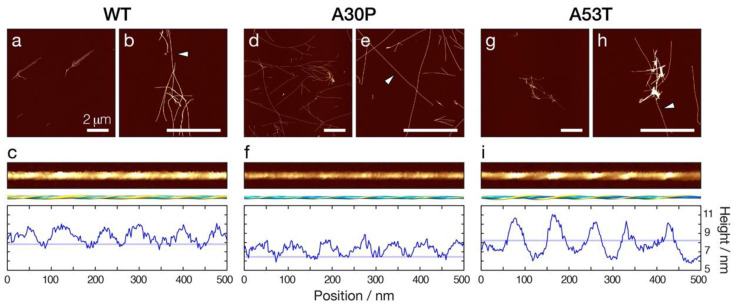
WT, A30P, and A53T α-synuclein amyloid fibrils formed in vitro display different fibril morphologies. AFM images of amyloid fibrils samples formed from WT (**a**,**b**), A30P (**d**,**e**) and A53T (**g**,**h**) α-synuclein. All three variants of α-synuclein amyloid fibrils were formed at 50 µM monomer equivalent concentration by gentle shaking at 180 rpm, 37 °C, for at least 14 days. AFM images representing 10 × 10 µm (**a**,**d**,**g**) or 4 × 4 µm (**b**,**e**,**h**) surface areas are shown with the scale bars indicating 2 µm. The white arrowheads in panels (**b**,**e**) and (**h**) indicate individual fibrils that are further magnified and shown in panels (**c**,**f**,**i**), respectively. In panels (**c**,**f**,**i**), the digitally straightened fibril images are shown together with 3D-reconstructed surface envelope models, their central-line height profiles (blue lines) and the population average height (pale blue lines) to demonstrate differences in fibril morphology.

**Figure 2 biomolecules-12-00630-f002:**
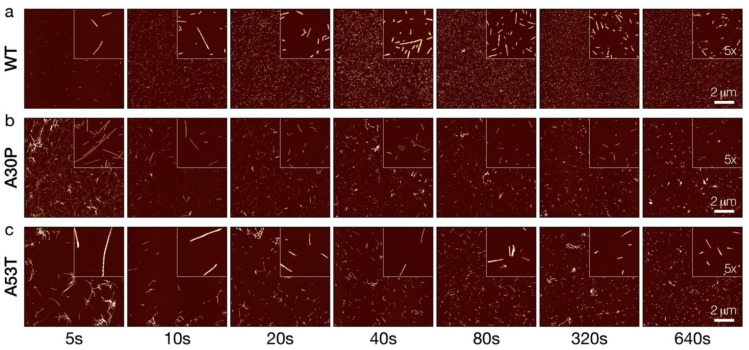
AFM imaging of WT, A30P and A53T α-synuclein amyloid fibrils undergoing fragmentation by controlled sonication. WT (**a**), A30P (**b**) and A53T (**c**) α-synuclein amyloid fibril samples formed from 50 µM monomer equivalent protein concentration were sonicated for up to 640 s, with the duration of sonication indicated for each time-point sample. AFM images representing 10 × 10 µm surface areas are shown with insets of 5× magnified images. The scale bars indicate 2 µm in all images.

**Figure 3 biomolecules-12-00630-f003:**
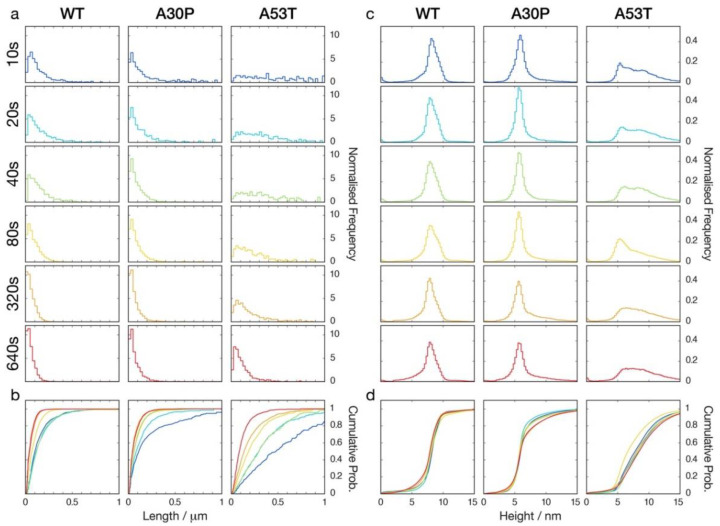
Fibril length and height distributions extracted from AFM images of the WT, A30P and A53T α-synuclein amyloid fibrils undergoing fragmentation by controlled sonication. Histograms representing normalised fibril length distributions (**a**) and height distributions (**c**) are shown using the same length and height scales, respectively, for comparison. Cumulative distribution functions of the same length distributions (**b**) and height distributions (**d**) are also shown to facilitate visualisation of the changes in the length distributions and the consistency of the height distributions for the duration of the controlled sonication. Statistics of the quantitative image analysis are shown in Appendix A.

**Figure 4 biomolecules-12-00630-f004:**
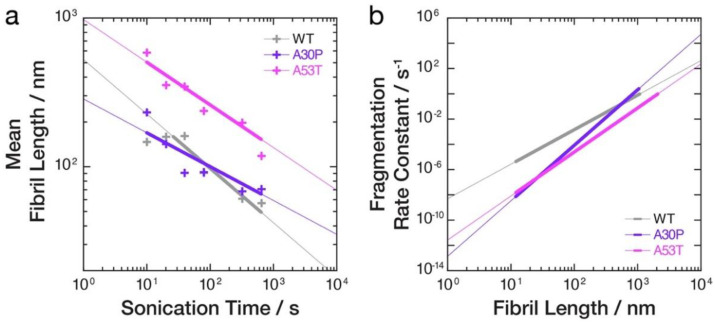
Comparative analysis of the relative stabilities of WT, A30P and A53T α-synuclein amyloid fibrils toward sonication-induced fragmentation. (**a**) The decay of experimentally observed mean lengths (+) as a function of sonication time duration is shown in a log–log plot together with the best-fit pure fragmentation model lines [26]. The thicker portion of the lines denotes the time ranges where the best-fit asymptotic lines describing characteristic length decay have likely been reached in the imaging experiments (the time ranges where the best-fit pure fragmentation model is likely reliable in describing the experimentally observed length decay, *t > t_s_* in Equation (1)). (**b**) The fragmentation rate constants, *B*(*x*), obtained from the best-fit pure fragmentation models are shown in a log–log plot as functions of fibril lengths, *x*. The thicker portion of the lines denotes the range of fibril lengths observed experimentally on the AFM images.

**Table 1 biomolecules-12-00630-t001:** Parameters describing the best-fit pure fibril fragmentation model to the sonication-induced length decay data (Figure 3). The *γ* parameter describing how the fragmentation rate constants depend on fibril length and the *α* parameter describing the typical magnitude of the division rate constants are shown with the length-dependent division rate constants *B*(*x*) *=* (*α·x*)*^γ^ s*^−1^ [26,27] at fibril length *x* = 100 nm (*B_100_*). The parameters are shown with their respective standard errors (SE) estimated as described in the Methods section. The average heights of the fibril particles throughout the sonication procedure are also shown with their standard error of mean (SEM).

	*γ* (±SE)	*α*/nm^−1^ (log *α* ± SE)	*B_100_*/s^−1^ (log *B_100_* ± SE)	Mean Fibril Height (±SEM)/nm
WT	2.7 ± 0.4	9.1·10^−4^ (−3.0 ± 0.1)	140·10^−5^ (−2.8 ± 0.2)	7.8 ± 0.1
A30P	4.4 ± 0.9	12·10^−4^ (−2.9 ± 0.1)	8.2·10^−5^ (−4.1 ± 0.6)	6.4 ± 0.1
A53T	3.4 ± 0.5	4.8·10^−4^ (−3.3 ± 0.1)	2.5·10^−5^ (−4.6 ± 0.5)	8.2 ± 0.2

## Data Availability

The article includes all datasets generated and analysed during this study. The list of raw fibril length distributions is available from the corresponding author on request.

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
