# Peer review of "Comparative Analysis of the Relative Fragmentation Stabilities of Polymorphic Alpha-Synuclein Amyloid Fibrils"

_biomolecules, 2022, doi:10.3390/biom12050630_

Round 1
Reviewer 1 Report
In this work, Sanami et al. present a comparative study of the mechanical stability of fibrils sampled from 3 distinct aSyn fibril preparations. Using AFM, they show that the 3 preps contain fibrils that are mophologically different, suggestive of different polymorphs. Using controlled sonication, they show that the fibrils from these preps are endowed with distinct mechanical stabilities, probably due to their different propensities to spontaneously form bundles by lateral assembly.
As highlighted by the authors, fibril fragmentation is a very important step in the process of amyloid spread. However, contrary to what is stated in the abstract and in the introduction, the possibility that different aSyn fibril polymorphs could present different fragmentation stabilities was previously explored. Using nanoparticle tracking analysis, it was indeed shown in 2020 that 2 different aSyn fibril polymorphs could yield different fragmentation patterns under the effect of sonication (DOI: 10.1126/sciadv.abc4364). Further, this difference was shown to translate into different sedimentation patterns in the presence of detergent, but perhaps most importantly, into different seeding and spreading acitivities in neurons. Thus, statements like "despite the mechanistic imortance of fibril fragmentation, the relative stabilities of different types or different polymorphs of amyloid fibrils toward fragmentation remains to be elucidated" (abstract) or "the relative fragmentation stability differences of distincts amyloid fibril polymorphs are not known" (introduction) are incorrect. Previous work on the very topic of the MS should be acknowledged.
Regarding the experimental approach of the MS I have a question regarding the definition of aSyn fibril polymoprphism based on Fig. 1: considerations are done here based on the longitudinal scans of 3 single fibrils, 1 fibril per fibril batch. Could the authors show series of such scans and do some comparative statistics? It would be important because aSyn fibril preps are very often mixtures of fibrils presenting different morphologies.
The same kind of question holds true for fig 3: although it is stated in the methods that the biases for fibril deposition on the mica surfaces were taken into account, it is difficult to figure out how many fibrils or fibril fragments were counted here, and how the mica surface was sampled to generate the histograms. Could this be displayed with statistics?
Regarding the best-fit pure fragmentation model previously developped by the authors and used in this MS, it would be interesting to have here a description intended for non specialist readers with an explanation of the assumptions underlying the model.
As to sup35 and HSP104, note that the first prosposal that HSP104 could promote yeast prion replication by causing fibril fragmentation can be found here DOI: 10.1016/s0092-8674(00)81216-7
In conclusion, this MS is a very interesting one and points to key mechanisms but deserves to be strengthened.
Author Response
Reviewer #1
As highlighted by the authors, fibril fragmentation is a very important step in the process of amyloid spread. However, contrary to what is stated in the abstract and in the introduction, the possibility that different aSyn fibril polymorphs could present different fragmentation stabilities was previously explored. Using nanoparticle tracking analysis, it was indeed shown in 2020 that 2 different aSyn fibril polymorphs could yield different fragmentation patterns under the effect of sonication (DOI: 10.1126/sciadv.abc4364). Further, this difference was shown to translate into different sedimentation patterns in the presence of detergent, but perhaps most importantly, into different seeding and spreading acitivities in neurons. Thus, statements like "despite the mechanistic imortance of fibril fragmentation, the relative stabilities of different types or different polymorphs of amyloid fibrils toward fragmentation remains to be elucidated" (abstract) or "the relative fragmentation stability differences of distincts amyloid fibril polymorphs are not known" (introduction) are incorrect. Previous work on the very topic of the MS should be acknowledged.
Reply: We apologise for the imprecise wording in the two statements pointed out by the reviewer. We have now modified these two sentences to emphasise on the need for quantification of the relative fragmentation stability differences. We have also added the reference suggested by the reviewer.
Regarding the experimental approach of the MS I have a question regarding the definition of aSyn fibril polymoprphism based on Fig. 1: considerations are done here based on the longitudinal scans of 3 single fibrils, 1 fibril per fibril batch. Could the authors show series of such scans and do some comparative statistics? It would be important because aSyn fibril preps are very often mixtures of fibrils presenting different morphologies.
The same kind of question holds true for fig 3: although it is stated in the methods that the biases for fibril deposition on the mica surfaces were taken into account, it is difficult to figure out how many fibrils or fibril fragments were counted here, and how the mica surface was sampled to generate the histograms. Could this be displayed with statistics?
Reply: This is an excellent suggestion. We have now added an additional Supplementary Table S1 describing the size of the dataset as well as the detailed image analysis statistics. This dataset is focused on the quantification of fibril particle dimensions so the image resolution was optimised for wide views with large coverage rather than higher resolution for detailed structural reconstruction of individual particles. Therefore, in-depth analysis of the polymorph distributions suggested by the referee (as exemplified by [25] in the original manuscript) is not possible using this dataset. However, the large number of particles analysed and the considerable shape differences in the height distributions confirms that the polymorph distributions are indeed different. To clarify on this point, we have modified the sentence on line 127 of the original manuscript.
Regarding the best-fit pure fragmentation model previously developped by the authors and used in this MS, it would be interesting to have here a description intended for non specialist readers with an explanation of the assumptions underlying the model.
Reply: We have now added a brief description of the model used to analyse the data in the Materials and Methods section.
As to sup35 and HSP104, note that the first prosposal that HSP104 could promote yeast prion replication by causing fibril fragmentation can be found here DOI: 10.1016/s0092- 8674(00)81216-7
Reply: We have added this additional reference suggested by the reviewer in addition to the very recent report that we originally cited ([19] in the original manuscript).
Reviewer 2 Report
This manuscript described using controlled sonication to study the fragmentation stability of three alpha-synuclein amyloid fibrils. First, they used AFM images to characterize the morphological differences of the three strains. Next, they showed that the A30P and A53T fibrils appeared to be more resistant to sonication-induced fragmentation. Data from image analysis provide quantitative evidence that fragmentation changed the length profiles drastically while keeping the height profiles unchanged. They also used a previously developed fragmentation model to fit the data and explain the observed variations. The experiments were well-designed and presented, supporting the claims the authors proposed nicely. I am excited to see the research community use this method in studying other amyloid fibrils. However, a few points should be addressed to increase the readability of this paper.
- 1a and b: a 2x magnification does not help visualize the fibrils that much. Could the authors provide 5x or 10x images?
- 1c, f, i: The authors should consider adding a line representing the average height because it is discussed in the text. Also, it would be nice to have cross-section (perpendicular to the fibril length axis) apparent height profiles of the three fibrils.
- 4: Please use colors with higher contrast to present the A30P and A53T data. Also, it would be helpful to include the values of fitted parameters in the figure panels.
- Is there a parameter to quantify the “clustering” of fibrils in the AFM images?
Author Response
Reviewer #2
- 1a and b: a 2x magnification does not help visualize the fibrils that much. Could the authors provide 5x or 10x images?
Reply: Presumably, the reviewer refer to the insets in Fig. 2. We have now replaced these 2x magnified insets with 5x magnified insets.
- 1c, f, i: The authors should consider adding a line representing the average height because it is discussed in the text. Also, it would be nice to have cross-section (perpendicular to the fibril length axis) apparent height profiles of the three fibrils.
Reply: We have added the population average line in Fig 1c, f and i. The cross-section scan-lines are heavily distorted by the tip-sample convolution effect (as discussed in [24] in the original manuscript). Therefore, we believe they do not add additional information compared to the height profile along the filament already shown.
- 4: Please use colors with higher contrast to present the A30P and A53T data. Also, it would be helpful to include the values of fitted parameters in the figure panels.
Reply: We have adjusted the colours and the symbol sizes to make the data points and fitted model lines more visible. The fitted parameters are already shown in table 1.
- Is there a parameter to quantify the “clustering” of fibrils in the AFM images?
Reply: We agree that a method to directly quantify fibril clustering represent an important challenge to be developed in the future.
Reviewer 3 Report
The submitted manuscript deals with the length-scale investigation of WT α-synuclein and its A30P and A53T variants with and without sonication. The different time-dependent samples showed that sonication induces distinct length-scale structures with some mathematical modeling-based calculations affording formation kinetics parameters. The authors do report useful AFM data but there are a few items that will need to be addressed before the manuscript can be accepted for publication. See comments below.
- The currently submitted manuscript by Sanami et al., does add to the investigation into α-synuclein fragmentation formation and analysis work done by Tarutani et al. published in 2016. In that work, researchers reported the formation of fibrillary higher-order structures formed by α-synuclein (shake-prepared at 37 ℃). Tarutani and co-workers also demonstrated that sonication time increase (0-180 sec) correlated with shorter fragment formations. Having said that it would be great to have some procedural and results comparison of this previous work (reference 30 in the manuscript) and the submitted manuscript. Also, do the size scales of the fragmentation matter physiologically? There's no testing done w/ cells or animals.
- The point mutations of α-synuclein have been investigated with and without sonication by Ruggeri and co-worked in their ACS Nano 2020 (v.14, pp.5213-5222) paper. They reported five different mutations including A30P and A53T and the variants all formed fibrillary structures analyzed via AFM. In the ACS Nano paper, WT, A30P, and A53T were grouped to having similar heights and rate of lipid induced aggregation plus secondary structure difference %. Quote: "The A53T and A30P variants show similar kinetic constants, structure, and morphology when compared to the WT protein..." So, how much different is the currently submitted work vs. what Ruggeri et al. have already reported in 2020?
- Also, based on what Ruggeri and colleagues reported, why weren't the variants E46K, H50Q, and G51D also tested, and compared to A30P, A53T, and the WT, as their morphological features are quite different? The comparisons among WT, A30P, A53T, E46K, H50Q, and G51D will be useful and helpful.
- Why are the AFM dilutions different? WT is 1 μM, A30P is 5 μM, and A53T is 25 μM. And then, for the non-sonicated or 5 sec sonicated samples only A30P was diluted to 25 μM. Why is the concentration independent variable different? In order for the AFM imaging experimental results to be compared, the concentrations should have been kept the same.
- The utilization of a probe sonicator (i.e., Qsonica Q124, 20 kHz probe sonicator w/ 2 mm probe) is one of the very important procedural details. This reviewer can understand that the amplitude is set to 20 %. Since the sonication is an important independent variable, how did the specific operating parameters get chosen? There are no other amplitude values or pulse conditions reported. Also, the reported data include 5, 10, 20, 40, 80, 320, 640 seconds (i.e., 5, 5*2, 5*2^2, 5*2^3, 5*2^4, 5*2^6, 5*2^7 sec) - where are the 5*2^5 = 160 sec data?
- It would be great to also see an increased discussion about the sonication process, which is bubble implosions at the μsec-msec time domain and μm-mm length scales - there are not enough details provided. What are potential effects that other researchers and scientists need to consider? Is there a temperature increase before and after sonication? Is there mechanically induced damage to the overall protein structure? Basically, can the authors provide more discussions about how the "controlled sonication" was achieved and how that tool/technology enabled, for instance, "controlled sonication promotes rapid fragmentation of α-synuclein amyloid fibrils?" You may want to review the following article: Nakajima et al., "Optimized sonoreactor for accelerative amyloid-fibril assays through enhancement of primary nucleation and fragmentation," Ultrasonic Sonochemistry, 2021, 73, 105508.
- Lines 189-190 and 306: The "Sticky" surface mentions should be defined more definitively or quantitatively. How would switching alanine to proline or threonine change the surface properties of the α-synuclein protein molecule and the fibrils it produces? Do you have surface charge information such as zeta potential measurements?
- Table 1: The fibril dimensions plus formation kinetics parameters, are they statistically different: WT vs. A30P vs. A53T? I do not see any inferential statistics performed? Also, how were the gamma parameters, alpha parameters, and B100 values calculated? The authors do cite their two papers as references 21 and 30. However, if the authors intend to keep their paper self-sustaining, they should strongly consider expanding on the explanation of the modeling/mathematic details, to a certain extent in the manuscript.
- Matlab M-file uploaded to Github, when executed, shows the following error message. The file was uploaded 7 months ago so perhaps the authors need to update with a more in-depth explanation since, after all, the script file is described as a tutorial but with limited information on how to use it by an average reader (who will most likely not know what Matlab is). Perhaps, the authors can explain the tutorial in the main text of the article. That will provide a unique opportunity for the readers to see how they can actually use AFM-derived information and get the parameter data as shown in Table 1. That will make this paper different than the authors' previous publications.
>>tutorial_DETX
Undefined function or variable 'moments'.
Error in tutorial_DETX>DETX_simu_inverse_paper (line 642)
Numb(i)=moments(X,u{i},0); %% Numb= total number of particles
Error in tutorial_DETX (line 200)
[gamma_e,alpha_e,Te,C]=DETX_simu_inverse_paper(X,u,time,N,1,0);
Author Response
Reviewer #3
- The currently submitted manuscript by Sanami et al., does add to the investigation into α- synuclein fragmentation formation and analysis work done by Tarutani et al. published in 2016. In that work, researchers reported the formation of fibrillary higher-order structures formed by α-synuclein (shake-prepared at 37 °C). Tarutani and co-workers also demonstrated that sonication time increase (0-180 sec) correlated with shorter fragment formations. Having said that it would be great to have some procedural and results comparison of this previous work (reference 30 in the manuscript) and the submitted manuscript. Also, do the size scales of the fragmentation matter physiologically? There's no testing done w/ cells or animals.
Reply: Indeed as the reviewer suggested, Tarutani et al. reported effects of sonicated alpha-synuclein fibrils in cellular and mice models. That is exactly the reason we included this interesting paper in our original manuscript (reference [31] in the original manuscript). In that study, they used a probe sonicator to generate fibril fragments. Therefore, we have now also added this reference in discussing methods of sonication (line 281 in the original manuscript) to facilitate procedural comparisons as the referee suggested.
- The point mutations of α-synuclein have been investigated with and without sonication by Ruggeri and co-worked in their ACS Nano 2020 (v.14, pp.5213-5222) paper. They reported five different mutations including A30P and A53T and the variants all formed fibrillary structures analyzed via AFM. In the ACS Nano paper, WT, A30P, and A53T were grouped to having similar heights and rate of lipid induced aggregation plus secondary structure difference %. Quote: "The A53T and A30P variants show similar kinetic constants, structure, and morphology when compared to the WT protein..." So, how much different is the currently submitted work vs. what Ruggeri et al. have already reported in 2020?
Reply: The work by Ruggeri et al. (reference [12] in the original manuscript) was focused on the morphology of the fibrils. Here, we report on the quantitative comparison of the fibrils’ fragmentation properties.
- Also, based on what Ruggeri and colleagues reported, why weren't the variants E46K, H50Q, and G51D also tested, and compared to A30P, A53T, and the WT, as their morphological features are quite different? The comparisons among WT, A30P, A53T, E46K, H50Q, and G51D will be useful and helpful.
Reply: A30P, A53T studied here are the two variants that showed significant effects with regard to both amplification and lipid induced aggregation compared to the other variants mentioned by the referee (Flagmeier et al., reference [11] in the original manuscript). However, we completely agree with the referee that following the data in the proof of concept study reported here, it would be very interesting indeed to map the fragmentation stability of a larger set of alpha-synuclein variants as well as other amyloid systems in the future.
- Why are the AFM dilutions different? WT is 1 μM, A30P is 5 μM, and A53T is 25 μM. And then, for the non-sonicated or 5 sec sonicated samples only A30P was diluted to 25 μM. Why is the concentration independent variable different? In order for the AFM imaging experimental results to be compared, the concentrations should have been kept the same.
Reply: The initial monomer equivalent concentration of all three variant samples were identical at 50μM. Therefore, the result can be compared since the concentration is the same during the sonication / fragmentation reaction. The concentrations used for AFM imaging were optimised so that the images obtained have good coverage but still contain well-separated and evenly dispersed fibril particles for quantitative analysis. We kept these AFM specimen deposition concentrations constant from 10s sonication and onwards for each dataset to enable analysis of the datasets as those are the time-points used in the analysis for each variant. We have modified the relevant section in the Materials and Methods (from line 376 in the original manuscript) to clarify on these points.
- The utilization of a probe sonicator (i.e., Qsonica Q124, 20 kHz probe sonicator w/ 2 mm probe) is one of the very important procedural details. This reviewer can understand that the amplitude is set to 20 %. Since the sonication is an important independent variable, how did the specific operating parameters get chosen? There are no other amplitude values or pulse conditions reported. Also, the reported data include 5, 10, 20, 40, 80, 320, 640 seconds (i.e., 5, 5*2, 5*2^2, 5*2^3, 5*2^4, 5*2^6, 5*2^7 sec) - where are the 5*2^5 = 160 sec data?
Reply: We chose the lowest possible amplitude on the sonicator because sonication provides a strong mechanical perturbation. This is now clarified in the Materials and Methods section. With regards to the chosen time-points, 160 second time-points of sonication were omitted in these experiments as a result of experimental optimisation involving sample consumption and microscopy usage considerations while retaining the advantage of having a long duration sonication time-point (i.e. 640 sec) for the analysis.
- It would be great to also see an increased discussion about the sonication process, which is bubble implosions at the μsec-msec time domain and μm-mm length scales - there are not enough details provided. What are potential effects that other researchers and scientists need to consider? Is there a temperature increase before and after sonication? Is there mechanically induced damage to the overall protein structure? Basically, can the authors provide more discussions about how the "controlled sonication" was achieved and how that tool/technology enabled, for instance, "controlled sonication promotes rapid fragmentation of α-synuclein amyloid fibrils?" You may want to review the following article: Nakajima et al., "Optimized sonoreactor for accelerative amyloid-fibril assays through enhancement of primary nucleation and fragmentation," Ultrasonic Sonochemistry, 2021, 73, 105508.
Reply: We have now added clarification of the experimental choices for the sonication procedure in the Materials and Methods section, including clarifying that the sonication was performed on ice to offset the heating up of the solution. We have also added the reference suggested by the reviewer in the discussions section in addition to the references on the mechanical mode of action of sonication (e.g. Huang et al. reference [27] of the original manuscript).
- Lines 189-190 and 306: The "Sticky" surface mentions should be defined more definitively or quantitatively. How would switching alanine to proline or threonine change the surface properties of the α-synuclein protein molecule and the fibrils it produces? Do you have surface charge information such as zeta potential measurements?
Reply: We already defined the clustering in terms of the right side tails of the height distributions (line 186 in the original manuscript). As already pointed out in response to referee 2, we do agree that a method to fully and directly quantify fibril clustering and fibril surface properties, either by image quantification or experimentally by zeta potential measurements represent an important challenge to be developed in the future.
- Table 1: The fibril dimensions plus formation kinetics parameters, are they statistically different: WT vs. A30P vs. A53T? I do not see any inferential statistics performed? Also, how were the gamma parameters, alpha parameters, and B100 values calculated? The authors do cite their two papers as references 21 and 30. However, if the authors intend to keep their paper self-sustaining, they should strongly consider expanding on the explanation of the modeling/mathematic details, to a certain extent in the manuscript.
Reply: We have already provided the standard errors of the estimated parameters in Table 1 and how the standard errors were estimated in the Materials and Methods section in the original manuscript. We have now modified the table legend to clarify on this. We have also extended the Materials and Methods section to include the model equations used to analyse the data.
- Matlab M-file uploaded to Github, when executed, shows the following error message. The file was uploaded 7 months ago so perhaps the authors need to update with a more in-depth explanation since, after all, the script file is described as a tutorial but with limited information on how to use it by an average reader (who will most likely not know what Matlab is).
Perhaps, the authors can explain the tutorial in the main text of the article. That will provide a unique opportunity for the readers to see how they can actually use AFM-derived information and get the parameter data as shown in Table 1. That will make this paper different than the authors' previous publications.
>>tutorial_DETX
Undefined function or variable 'moments'.
Error in tutorial_DETX>DETX_simu_inverse_paper (line 642)
Numb(i)=moments(X,u{i},0); %% Numb= total number of particles Error in tutorial_DETX (line 200)
[gamma_e,alpha_e,Te,C]=DETX_simu_inverse_paper(X,u,time,N,1,0);
Reply: We apologise for the inconsistencies in the tutorial code file, which was originally written for an mathematical/computational biology audience. As the referee pointed out, it assumes the reader knows the moments of distributions functions (e.g. the first moment being the mean, which is used here in the data analysis) and other mathematical concepts etc. We have now replaced this sentence with an extended section to include the model equations used to analyse the data and a brief description of the analysis workflow. Which we hope would be more accessible to wider audience.
Round 2
Reviewer 1 Report
The Authors have clearly improved their manuscript.
However lines 67 and 68 now read:
"The relative fragmentation stability differences of distinct amyloid
fibril polymorphs have not been quantified, but...."
As detailed in my previous review, it should instead read:
"The relative fragmentation stability differences of distinct α-synuclein
fibril polymorphs revealed by sonication has previously been documented [10.1126/sciadv.abc4364] but was not quantified...
This point should be corrected.
Author Response
We thank the reviewer for their suggestion for the sentence on lines 67-68 and we have now edited the wording as suggested.
Reviewer 3 Report
Thank you for updating/revising the manuscript. Appreciate including the mathematical equations and making other adjustments such as providing additional context (i.e., Table S1) to how controlled sonication contributes to fibril fragmentation. The revisions, overall, do appear to offer better details that will help other scientists reproduce the describe experiments. With that note, I do feel that the Matlab example/tutorial on Github is still useful to make aware for the readers. Please do consider including the URL (below) and perhaps explain some of the sections in the supporting info. I do see the usefulness of introducing the readers of Biomolecules on how Matlab can help model sub-micron-sized events such as amyloid fibril fragmentation.
https://github.com/mtournus/Fragmentation
Author Response
We thank the reviewer for their support. As suggested by the reviewer, we have now added the url to the Matlab code examples back into the manuscript in the hope that it will help readers to test the approach reported here on their own amyloid samples.